# Syntheses of Colorless and Transparent Polyimide Membranes for Microfiltration

**DOI:** 10.3390/polym12071610

**Published:** 2020-07-20

**Authors:** Jong Won Kim, Jin-Hae Chang

**Affiliations:** Department of Polymer Science and Engineering, Kumoh National Institute of Technology, Gumi 39177, Korea; kiw010@naver.com

**Keywords:** colorless and transparent membrane, polyimide, poly(vinyl alcohol), blend

## Abstract

Herein, poly(amic acid) (PAA) was synthesized using 4,4’-(hexafluoroisopropylidene) diphthalic anhydride (6FDA) as a dianhydride and 2,2-bis(3-aminophenyl)hexafluoropropane (6FAm) and 2,2-bis(3-amino-4-hydroxyphenyl)hexafluoropropane (6FAm-OH) as diamines. Poly(vinyl alcohol) (PVA) at various contents (0–5.0 wt%) was blended with PAA to prepare a composite material. Then, colorless and transparent polyimide (CPI) composite films were prepared by applying various stages of heat treatment using the PAA/PVA blend film as a precursor. These film-type composites were immersed in water to completely dissolve PVA, a water-soluble polymer, and their pore sizes were investigated to determine their potential as a porous membrane. According to the results of scanning electronic microscopy (SEM), as the concentration of PVA increased from 0 to 5.0 wt% in the CPI/PVA composite films, the size of the pores resulting from the dissolution of water-soluble PVA increased. Further, the micrometer-sized pores were uniformly dispersed in the CPI films. The thermal properties, morphology, and optical transparency of the two types of CPI membranes synthesized using 6FAm and 6FAm-OH monomers were examined and compared.

## 1. Introduction

Several impurities, such as synthetic materials, present in drinking water and industrial wastewater, are difficult to remove naturally due to their complex structures. Conventional methods for treating wastewater and contaminated solvents include the use of filters, biological drainage, ozonation, coagulation flocculation, powdered activated carbon adsorption, electrochemical processes, and photocatalysts [1,2,3,4]. However, it is difficult to completely remove impurities using these methods. In addition, most of the abovementioned methods result in the generation of new impurities due to the addition of chemicals during the purification process.

Membrane processes, including microfiltration (MF), ultrafiltration (UF), nanofiltration (NF), and reverse osmosis (RO), have been used for water purification and wastewater and dye treatment. These membrane processes were developed to separate materials of different sizes and types. That is, MF can separate particles, UF can separate macromolecules, and NF can separate nano-sized materials. In addition, RO can separate ionic compounds. Therefore, of the membrane processes used in current water treatment technologies, MF is employed to remove suspended solids, protozoa, and bacteria, and UF is used to remove viruses and colloids. NF is commercially available for the removal of hardness, heavy metals, and dissolved organic matter, and RO is used for desalination, water reuse, and water purification. These methods have already been widely applied in various field and are described in numerous studies [5,6,7]. However, in pressure-based liquid-phase membrane processes, flux reduction due to concentration polarization and fouling is a problem to overcome, to prevent additional resistance to the transport of materials through the membrane.

The properties and effects of several MF membranes produced for water treatment have been characterized. In 1997, Mueller et al. [8] reported that a ceramic MF membrane using polyacrylonitrile (PAN) achieved 99% purity in the treated water. Moreover, Zhong et al. [9] developed a next-generation ceramic MF membrane using zirconia (ZrO_2_), which has been proven to be more effective in water treatment. In addition, according to previous MF membrane studies, inorganic membranes such as ceramic membranes were more effective in reducing fouling via pre-treatment stages such as flocculation [10].

Research results [11,12] have been applied to the production of inorganic UF and NF membranes since the 1940s. In general, an inorganic membrane has a dense structure on a porous support. Materials used to produce inorganic membranes include mechanically stable alumina, silica, silver, zirconia, mullite, oxide mixtures such as titanium oxide (TiO_2_) and zinc oxide (ZnO), and sintered metal. In particular, by using nano-sized particles, NF membranes can be manufactured with enhanced selective permeability, mechanical properties, and hydrophilicity. Zhang et al. [13] used the inorganic precursors tetraethoxysilane (TEOS) and tetra-n-butyl titanate (TBOT) in polyethyleneimine (PEI) to prepare PEI-silica and PEI-titania nanocomposite membranes, respectively. These NF membranes exhibited excellent thermal properties and were structurally stable [14].

Unlike the membrane technologies described above, photocatalytic technology uses chemical reactions initiated by light and is a useful technology capable of producing hydrogen energy and purifying air and water without environmental load. Currently, it is receiving attention as a way to solve serious energy and environmental problems. In particular, photocatalytic membrane reactors (PMRs) using photocatalytic technologies are inexpensive and environmentally friendly and have been shown to be highly efficient in the treatment of water and wastewater [15]. In addition, photocatalytic reactors can decompose untreated organic and toxic pollutants present in water sources and treated sewage effluent into simple, harmless inorganic molecules. PMRs are superior to conventional photocatalytic reactors in that it is possible (1) to reduce the loss of photocatalysts in large-scale applications, (2) to control the retention time of molecules in the reactor, and (3) to continuously separate catalysts and products. Other advantages of PMRs include improved process efficiency and stability, and reduced cost by reusing photocatalysts [16].

Polyimides (PIs) can be easily processed into the form of a membrane with high chemical and thermal stability over a wide range of operating conditions. PI membranes are known to have excellent permeability for gas or liquid phase separation [17,18]. As a conventional polymeric material, PI has extensively shown its commercial potential as a permeable membrane material with good physicochemical properties, including high thermal stability, excellent mechanical properties, and good processability [19]. PI can also interact with water molecules or smaller molecules because the imide groups of PI can form hydrogen bonds with these molecules. Despite the abovementioned advantages, most of the PI membranes suffer from stability problems due to expansion at high temperatures, resulting in reduced production stability and poor performance during the separation process. Therefore, various methods, such as blending, heat treatment, and crosslinking, have been developed for the modification of PI membranes to achieve improved stability, performance, and physicochemical properties. Among them, blending has been widely used as a simple and inexpensive method to improve the original properties of PI membranes [20,21].

Generally, PI is dark brown due to the charge transfer complex (CT complex) formed between the straight and rigid imide and benzene structure [22,23,24]. The formation of this CT complex can be inhibited by introducing a bent monomer with a strong electron-withdrawing group such as trifluoromethyl (–CF_3_) and sulfone (–SO_2_–) into the PI main chain, thereby hindering the movement of electrons between the PI backbones. Thus, by simply modifying the structure of the monomer, optically colorless and transparent PI (CPI) can be synthesized. CPIs containing the –CF_3_ group exhibit a high modulus and low coefficient of thermal expansion and can be used in electronic materials such as flexible display materials and transparent electrodes [25,26]. As described above, the main chain of CPI is a kinked structure and includes an asymmetrical substituent. Thus, the processing temperature is lower and the membrane manufacturing process is superior to that of conventional PI. In addition, conventional PI is dark yellow or brown, so it is mainly used as an interior material. Although its thermal stability is slightly inferior to conventional PI, CPI is colorless and transparent. Therefore, it can be used for interior materials and also for exterior materials that require optical transparency. 

Poly(vinyl alcohol) (PVA) is a water-soluble polymer extensively used in paper coating, textile sizing, and production of flexible water-soluble packaging films [27]. Such applications have stimulated interest in improving the mechanical, thermal, and permeability properties of thin film composite (TFC) membranes [11,28,29,30]. Since PVA exhibits strong water solubility, it is possible to make pores of uniform size when preparing a TFC membrane in aqueous solution. This result allows the membrane to maintain uniform properties.

Polymers used in various types of MF and UF membranes include PAN, PVA, polyether sulfone (PES), polysulfone (PSF), polypropylene (PP), polytetrafluoroethylene (PTFE), poly (vinylidene fluoride) (PVDF), and sulfonated PSF and PES. These polymer membranes show excellent selectivity, stability, and permeability during water treatment. PES and PSF membranes are not only the most used materials for UF membranes, but are also widely used for NF and RO composite membranes. In addition, PP and PVDF are also widely used materials for MF membranes [10,12].

Herein, we synthesized two poly(amic acid)s (PAAs) by reacting 4,4’-(hexafluoroisopropylidene) diphthalic anhydride (6FDA) as a dianhydride with 2,2-bis(3-aminophenyl) hexafluoropropane (6FAm) and 2,2-bis(3-amino-4-hydroxyphenyl)hexafluoropropane (6FAm-OH) as diamines. The blend materials were fabricated by blending PVA in various amounts, ranging from 0 to 5 wt% with the synthesized PAAs. The CPI/PVA blend films were prepared by heat treatment at various temperatures, and porous CPI membranes were prepared after removing PVA from the films. The thermal properties, morphologies, and optical transparencies of the two types of CPI films, synthesized using two different monomers (6FAm and 6FAm-OH), were investigated and compared.

The purpose of this study was to investigate the possibility of synthesizing membranes using CPI in solution blending processes and to determine the suitability of CPI for the manufacture of membranes by characterizing their thermal and optical properties. Further, we aimed to control the pore size in CPI membranes by a simple solution-based process, such as the dissolution of a water-soluble filler. 

## 2. Experimental Details

### 2.1. Materials

In this study, 6FAm was purchased from Santa Cruz Biotechnology (Shanghai, China) and 6FAm-OH was purchased from TCI (Tokyo, Japan). DMAc, a solvent, was purchased from Junsei Chemical Co. (Tokyo, Japan) and used after completely removing the moisture with a molecular sieve (4 Å). PVA with 80% saponification (DP = 2.04 × 10^2^) was purchased from Aldrich Chemical Co. (Yongin, Korea). and used as received. A general purpose solvent was used as it did not require purification.

### 2.2. Preparation of the CPI/PVA Blend Film

With each of the two monomers, 6FAm and 6FAm-OH, PI was synthesized by the same method. Therefore, the method is described for the synthesis using 6FAm: In a 250-mL three-necked flask, 4.39 g (1.3 × 10^−2^ mol) 6FDA was added to 80 mL of DMAc followed by stirring for 1 h for complete dissolution. Then, 5.83 g (1.3 × 10^−2^ mol) 6FAm was dissolved in this solution. Under a nitrogen atmosphere, a PAA solution was prepared by slowly stirring the abovementioned solution for 1 h at room temperature for stabilization, followed by stirring at 0 °C for 1 h and at room temperature for 12 h.

After mounting a reflux reactor on the three-necked flask, 0.10 g (2.27 × 10^−3^ mol) PVA corresponding to 1 wt% PAA solid was added to the abovementioned solution and dispersed by stirring for 3 h at 50 °C under a nitrogen atmosphere. The resulting PAA/PVA solution was evenly spread on a clean glass plate and cast, and then, the solvent was slowly removed while stabilizing PAA for 2 h at 50 °C in a vacuum oven. Thereafter, vacuum was applied at 80 °C for 1 h to remove the solvent.

The thermal imidization reaction system was maintained under a vacuum atmosphere at 110, 140, and 170 °C for 30 min, followed by heat treatment at 195 and 220 °C for 50 min, and finally, maintained at 235 °C for 2 h to complete thermal imidization. The prepared CPI/PVA film was slowly removed from the glass plate in water at 90 °C. The size of the films were 50 × 50 mm^2^. A detailed synthetic route is shown in Scheme 1. 

### 2.3. Preparation of a Porous CPI Membrane Film

PVA is a hydrophilic polymer that is completely soluble in water; thus, the CPI/PVA blend film was immersed in water at 30 °C for approximately 8 h to completely dissolve the PVA component in the blend film, and a porous CPI membrane film of a desired size (thickness: 18–21 μm) was obtained. Then, this porous CPI membrane film was dried in a vacuum oven at 50 °C for 12 h.

### 2.4. Characterization

Fourier-transform infrared spectroscopy (FTIR) (Bruker, VERTEX 80v, Berlin, Germany) was used to confirm the synthesis of the CPI membrane film, with particular attention to the absorption peak related to the imide group. Carbon-13 nuclear magnetic resonance (^13^C-NMR, Bruker 400 DSX NMR, Berlin, Germany) spectra were recorded at the Korea Basic Science Institute in Western Seoul Center.

Differential scanning calorimetry (DSC, NETZSCH 200F3, Berlin, Germany) and thermogravimetric analysis (TGA, TA instrument TA Q500, New Castle, DE, USA) were conducted under a nitrogen atmosphere to investigate the thermal properties of the CPI membrane film. The temperature increased at a rate of 20 °C/min. The degrees of crystallinity of the films were determined using an X-ray diffractometer (XRD, Rigaku SWXD, Tokyo, Japan), with Cu-Kα radiation. The measurement was conducted at a scanning speed of 2°/min in the range of 2θ = 2°–32°. The pore sizes of the films were examined using a field scanning electron microscope (SEM, JEOL JSM-6500F, Tokyo, Japan). To characterize the optical properties, a color difference meter (Konica Minolta CM-3600D, Tokyo, Japan) and an ultraviolet/visible spectrophotometer (UV-vis, Shimadzu UV-3600, Tokyo, Japan) were used. 

## 3. Results and Discussion

### 3.1. FTIR and NMR

Figure 1 shows the FTIR spectra of the CPI films containing 6FAm and 6FAm-OH. For the CPI films containing 6FAm (6FAm PI), the C=O aromatic stretching peaks were observed at 1787 and 1720 cm^−1^, and the C–N–C peak at 1368 cm^−1^ indicated the imidization of PI. By contrast, for the CPI films containing 6FAm-OH (6FAm-OH PI), a broad O–H stretching peak was observed around 3120 cm^−1^. As the -OH group is sensitive to hydrogen bonding, the intensity of the -OH stretching peak observed for this CPI film was less than that of the typical hydroxyl peak. Molecules with -OH functionalities are capable of forming intramolecular hydrogen bonds in the PI main chain and show broad stretching absorption peaks over a wide range of 3000–3300 cm^−1^. Therefore, the spectrum of 6FAm-OH PI in Figure 1 shows a broad peak for the hydrogen bond between the phenolic -OH group and nitrogen in the adjacent imide rings [31]. Similar to the case of 6FAm PI, the spectrum of 6FAm-OH PI shows C=O aromatic stretching peaks at 1786 and 1718 cm^−1^, and the C-N-C peak indicating imidization was observed at 1378 cm^−1^. These FTIR results suggest that both CPIs underwent a complete imidization reaction [32]. 

Solid-state NMR was used to further confirm the structure of the synthesized CPI films, and the results are shown in Figure 2. The chemical shifts of the carbons in 6FAm PI and 6FAm-OH PI were recorded by solid-state ^13^C CP/MAS NMR at room temperature. The MAS rate was set to 10–12 kHz, and the spinning sidebands are marked with an asterisk. For 6FAm PI, the ^13^C chemical shifts of the benzene ring were observed at 127.80, 132.06, and 137.41 ppm. The chemical shifts of ^13^C adjacent to –CF_3_, ^13^C in –CF_3_, and the carbonyl carbon were observed at 64.67, 127.80, and 165.23 ppm, respectively, as shown in Figure 2a. For 6FAm-OH PI, the chemical shifts of ^13^C in benzene were observed at 118.18, 132.77, and 137.03 ppm. Moreover, the chemical shifts of ^13^C adjacent to –CF_3_, ^13^C in –CF_3_, and ^13^C adjacent to –OH were observed at 63.96, 124.18, and 153.49 ppm, respectively, as shown in Figure 2b. The ^13^C signal at 165.94 ppm corresponds to the carbonyl carbon [33,34]. The chemical shifts of all carbons are consistent with the structures shown in Figure 2. 

### 3.2. Thermal Properties

The thermal properties of the CPI, PVA, and CPI membranes with various contents (wt%) of PVA in the CPI synthesized using two diamines, 6Fam and 6FAm-OH, are summarized in Table 1. The *T_g_*s of the CPI films containing 6FAm showed a nearly constant value at 233–238 °C, as the PVA content increased from 0 to 5.0 wt%. This result was the same for the CPIs containing 6FAm-OH. For example, 6FAm-OH PIs showed a constant *T_g_* value of 302–303 °C, regardless of the amount of PVA in the PI blend. Upon comparing the *T_g_* values of the CPI blends with 6FAm and 6FAm-OH, the *T_g_* values of the CPI membrane with 6FAm-OH were found to be higher than those of the CPI membrane with 6Fam, regardless of the amount of PVA. These results show that the -OH groups of 6FAm-OH in CPI increase the *T_g_* value of PI through hydrogen bonding, resulting in intermolecular attraction between PI chains. The hydrogen bonds between the CPI polymer chains impede free chain movement, thereby increasing *T_g_*. Similar results have been reported in numerous studies [35,36]. Figure 3 shows the DSC thermograms of the CPI membranes with various PVA contents. 

Table 1 also shows the initial decomposition temperature (*T_D_^i^*) of the CPI blend films containing 6FAm and 6FAm-OH, depending on the concentration of PVA. In the CPI membranes with 6FAm, although the PVA content increased from 0 to 5.0 wt%, their *T_D_^i^* values remained constant at 348–351 °C (see Table 1). This result was almost the same as that obtained for the CPI membranes containing 6FAm-OH; with an increase in the PVA content from 0 to 5.0 wt%, the *T_D_^i^* value remained constant at 327–329 °C. The TGA thermograms of the CPI membranes obtained using two different monomers are shown in Figure 4. For the two CPI membrane series, no transition temperatures (*T_g_*, *T_m_*, and *T_D_^i^*) corresponding to PVA were observed, and the thermal properties of each series were almost the same, regardless of the PVA content; this indicates that PVA was completely removed from the CPI blend and a pure CPI membrane was obtained.

In the TGA thermograms shown in Figure 4b for the CPI membrane film with 6FAm-OH, several stages of thermal decomposition were observed upon heating. This is because PI with the -OH group undergoes thermal rearrangement (TR) with polybenzoxazole (PBO) at high temperatures [37,38,39]. Several studies have demonstrated that PBO membranes produced by TR show unusual microporous properties due to a significant increase in free volume during the TR process in the solid phase. The PBO membranes obtained by the heat treatment of PI with -OH groups are excellent materials for gas separation applications, such as CO_2_ separation for carbon capture, because they exhibit excellent gas selectivity for mixed gases such as CO_2_/CH_4_ and CO_2_ /N_2_ [38,39]. The process of PBO formation by the heat treatment of PI containing an -OH group is shown in Scheme 2.

Similar to the values of *T_g_* and *T_D_^i^*, values of the weight residue at 600 °C (*wt_R_^600^*) for the two CPI membrane films remained almost constant, regardless of the PVA content. That is, the *wt_R_^600^* values of the CPI membrane films containing 6FAm and 6FAm-OH remained 73–77% and 55–58%, respectively, regardless of the concentration of PVA. Unlike *T_g_*, the *T_D_^i^* and *w_R_^600^* values of the PIs with 6FAm-OH are lower than those of the PIs with 6FAm due to the low thermal stability caused by the -OH groups in 6FAm-OH. The values of *T_g_*, *T_D_^i^,* and *wt_R_^600^* indicated that PVA exerted negligible effects on the thermal properties of the films because the PI membrane films were immersed in water to remove the water-soluble PVA. 

### 3.3. Morphology

Figure 5 shows the XRD patterns of the two types of CPI membrane films with PVA at various contents. The characteristic peak of PVA was found at 2θ = 19.68° (d = 4.51 Å) and 23.64° (d = 3.75 Å). However, for 6FAm PI, even when the concentration of PVA increased from 0 to 5.0 wt%, no characteristic peaks corresponding to PVA were observed, and a similar result was obtained for 6FAm-OH PI. Additionally, there was almost no change in the degree of crystallinity, regardless of the amount of PVA that was added to the PI membrane film (see Table 1). These results indicate that the PVA component was completely removed, regardless of the concentration of PVA in the PI blend films. The data obtained using XRD are the initial results on filler dispersion in the membrane film and must be cross-checked with an electron microscope to comprehensively observe the removal of the filler.

The morphology of the films was investigated by immersing them in water to dissolve the water-soluble PVA portion of the CPI/PVA blend films. SEM was used to confirm the porosity of the CPI membrane films in which PVA was removed. Figure 6 shows the SEM images of the porous CPI membrane films, with 6FAm containing PVA at varying concentrations. As the concentration of PVA increased from 0 to 5.0 wt%, the pore size gradually increased. For example, when the PVA concentration was 0.5 wt%, the average pore diameter was 0.14 μm; however, when the PVA concentration increased to 2.0 wt%, the pore diameter also increased to approximately 0.41 μm. The pore size increased to approximately 0.69 μm at the PVA concentration of 5.0 wt%. Most of the pores were evenly distributed throughout the PI film, as shown in Figure 6. 

As the PVA content of the CPI blend increased from 0.5 to 5.0 wt%, PVA agglomerated to increase the particle size. When these particles are dissolved in water and removed, the pore size in the CPI membrane correspondingly increased. The shape of the membrane pores, depending on the concentration of PVA, is also shown in Figure 6.

Similar results were obtained for CPI with 6FAm-OH, as listed in Table 1. As the amount of PVA in this CPI/PVA blend increased from 0 to 5.0 wt%, the pore diameter slowly increased. For example, when the amount of PVA in the CPI membrane increased from 0 to 1.0 wt%, the pore diameter increased to 0.13 um, and when the amount of PVA increased to 5.0 wt%, the diameter of the pores increased to 0.61 um on average. The variation in pore sizes with increasing PVA concentration in the CPI membrane films with 6FAm-OH is shown in Figure 7.

To investigate the overall pore size and dispersion of pores in the CPI membrane, SEM images of the two series of films containing the same concentration of PVA were obtained at a low magnification of ×3000, and the results are shown in Figure 8. The pore diameters of the CPI membrane films with 6FAm and 6FAm-OH containing 5 wt% PVA were approximately 0.64–0.75 and 0.53–0.69 μm, respectively; moreover, the overall size of the pores was constant and the dispersion was excellent.

Upon comparing the average diameters of the CPI membranes containing two different monomers, the average pore diameters of the CPI films containing 6FAm-OH were found to be smaller than those of the CPI films containing 6FAm at the same PVA content. This result is attributable to the formation of dense structures due to the hydrogen bonds between 6FAm-OH PI and PI and PVA containing -OH groups. PVA was then dissolved and removed from the PI blend film, and the removal of PVA can be verified by the smaller pores in the CPI film containing 6FAm-OH than those in the CPI film containing 6FAm, as shown in Table 1. 

### 3.4. Optical Transparency

The optical properties of the synthesized CPI membrane films can be represented by the cut-off wavelength (λ_o_), transmittance at 500 nm wavelength (500 nm^trans^), and yellow index (YI) [40,41]. The UV transmittances and summarized results are shown in Figure 9 and Table 2, respectively. It was found that the λ_o_ value, indicating the initial transmission in all CPI membrane films, was <400 nm, and the films started to transmit light before the visible-light region. CPI with 6FAm showed very low λ_o_ values of 280–290 nm regardless of the PVA content. However, CPI with 6FAm-OH showed a λ_o_ value of 345–360 nm at various PVA contents of 0–5 wt%. The λ_o_ values of CPI with 6FAm-OH were higher than those of CPI with 6FAm because the hydroxyl groups in 6FAm-OH were hydrogen-bonded to form a denser polymer structure, as explained earlier. However, these values are very low as compared to those of other polymer films and are indicative of the almost colorless and transparent properties of the films. 

The two series of CPI membrane films exhibited excellent optical properties, with a maximum UV transmittance of 87–89% at 500 nm regardless of the content of PVA, as shown in Table 2. The YI of the two series of CPI membrane films with different contents of PVA was measured, and the results are also listed in Table 2. The YI values of these CPI membrane films are very low (2–3). In the CPI/PVA blend, when PVA was removed from the aqueous solution, the YI value of the CPI blend remained constant, regardless of the PVA content. All CPI membrane films containing PVA in the range of 0–5.0 wt% were almost colorless and transparent. Additionally, the optical transparency was excellent because there was no difficulty in reading the letters through the film. Since the PVA component of the CPI membrane was completely removed, it was not possible to determine the effect of the PVA content on the optical transparency in the two membranes series, as demonstrated with the thermal properties of the films.

## 4. Conclusions

Colorless and transparent porous PI membranes were prepared using a CPI/PVA blend. To synthesize CPI, 6FDA was used as a dianhydride and 6FAm and 6FAm-OH were used as diamines. To prepare porous membranes, water-soluble PVA was used in the CPI blend. As 6FAm-OH contains -OH groups in the main chain, hydrogen bonding is possible not only in the PI chain itself but also between PI and PVA present in the blend. 

In this study, the CPI membranes synthesized using 6FAm and 6FAm-OH were compared with each other at different PVA concentrations. As a very thin CPI membrane film (18–21 μm) was prepared, a pure CPI membrane could be prepared by almost completely removing the PVA component in an aqueous solution. The *T_g_* of CPI with 6FAm-OH was higher than that of CPI with 6FAm PI due to hydrogen bonding between the polymer chains; in the latter films, the pore diameter was small due to the tighter structure. In contrast, the *T_D_^i^* and *wt_R_^600^* values of CPI with 6FAm-OH were lower than those of CPI with 6FAm due to the low thermal stability of the -OH group. Regardless of the amount of PVA, the optical transparency of all the obtained CPI membranes was excellent.

In conclusion, these CPI membrane films are expected to be useful as high functional polymeric materials as well as in the field of filtration, due to their superior thermal property and optical transparency as compared to that of general purpose engineering polymers. Specifically, in the case of 6FAm-OH PI, because of its hydrophilicity, the pore size can be easily adjusted. Therefore, it is expected that 6FAm-OH PI can be used for MF membranes that require optical transparency.

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
