# Peer review of "Syntheses of Colorless and Transparent Polyimide Membranes for Microfiltration"

_polymers, 2020, doi:10.3390/polym12071610_

Round 1

Reviewer 1 Report

This paper presented a colorless and transparent polyimide (CPI) composite membrane containing poly(amic acid) and poly(vinyl alcohol). The review comments for this paper are in the following:
1. More comparisons of various water treatment methods should be conducted in Introduction.
2. How is the status of wastewater treatment using the inorganic materials? Some related work should be mentioned (A review on reverse osmosis and nanofiltration membranes for water purification; Membrane fouling in photocatalytic membrane reactors (PMRs) for water and wastewater treatment: A critical review; etc.). What is the water purification efficiency of these methods?
3. What are other widely used polymeric materials for water purification?
4. The Introduction section should be improved. Some of the work in literature should be discussed in details especially the developments of various membranes for water treatment process.
5. In Table 1, please unify the pressure units. The temperature unit is incorrect. I feel like that these conditions are not necessary.
6. Did you use knife coating for the film to control the membrane thickness?
7. In Figure 1, the unit of wavenumber should be cm-1. Figure 2 should be placed in Section 3.1.
8. I suggest the authors to use the curves in the figures with different colors. For Figure 4, it is so hard to distinguish these lines. Why did the blended film containing 5 wt% PVA show the best thermal stability in Figure 4b?
9. Line 245, the PBO formation process should be listed as a scheme.
10. Figure 5 is in a very low quality.
11. Figures 10 and 11 are useless if all the films are transparency.
12. The conclusions should be shortened.

13. The English writing should be improved significantly.

14. Please improve the reference format in the revision.

Author Response

Dear Editor                                      

This is my response to your comments regarding our paper “Syntheses of Colorless and Transparent Polyimide Membranes for Microfiltration” (polymers-850893) in polymers.

Thank you very much for the referee's comments. I have carefully revised the manuscript following the comments of the referee.

Reviewer-1

Q1. More comparisons of various water treatment methods should be conducted in Introduction.

A1. As pointed out by the reviewer, various water treatment methods were described in the introduction and compared with each other. See page 1, 38th line.

“Membrane processes, including microfiltration (MF), ultrafiltration (UF), nanofiltration (NF), and reverse osmosis (RO), have been used for water purification and wastewater and dye treatment. These membrane processes were developed to separate materials of different sizes and types. That is, MF can separate particles, UF can separate macromolecules, and NF can separate nano-sized materials. In addition, RO can separate ionic compounds. Therefore, of the membrane processes used in current water treatment technologies, MF is employed to remove suspended solids, protozoa, and bacteria, and UF is used to remove viruses and colloids. NF is commercially available for the removal of hardness, heavy metals, and dissolved organic matter, and RO is used for desalination, water reuse, and water purification.”

Q 2-1. How is the status of wastewater treatment using the inorganic materials? Some related work should be mentioned (A review on reverse osmosis and nanofiltration membranes for water purification;

A 2-1. The content of the inorganic membrane was supplemented according to the reviewer's point. See page 2, 57th line. The following sentence was supplemented in the Introduction.

“Research results [11,12] have been applied to the production of inorganic UF and NF membranes since the 1940s. In general, an inorganic membrane has a dense structure on a porous support. Materials used to produce inorganic membranes include mechanically stable alumina, silica, silver, zirconia, mullite, oxide mixture such as titanium oxide (TiO2) and zinc oxide (ZnO), and sintered metal. In particular, by using nano-sized particles, NF membranes can be manufactured with enhanced selective permeability, mechanical properties, and hydrophilicity. Zhang et al. [13] used the inorganic precursors tetraethoxysilane (TEOS) and tetra-n-butyl titanate (TBOT) in polyethyleneimine (PEI) to prepare PEI-silica and PEI-titania nanocomposite membranes, respectively. These NF membranes exhibited excellent thermal properties and were structurally stable [14].

Q 2-2. Membrane fouling in photocatalytic membrane reactors (PMRs) for water and wastewater treatment: A critical review; etc.). What is the water purification efficiency of these methods?

A 2-2. The contents of PMR were supplemented in line 66 on page 2 according to the reviewer's point.

“Unlike the membrane technologies described above, photocatalytic technologies use chemical reactions initiated by light, and is a useful technology capable of producing hydrogen energy and purifying air and water without environmental load. Currently, it is receiving attention as a way to solve serious energy and environmental problems. In particular, photocatalytic membrane reactors (PMRs) using photocatalytic technologies are inexpensive and environmentally friendly and have been shown to be highly efficient in the treatment of water and wastewater [15]. In addition, photocatalytic reactors can decompose untreated organic and toxic pollutants present in water sources and treated sewage effluent into simple, harmless inorganic molecules. PMRs are superior to conventional photocatalytic reactors in that it is possible 1) to reduce the loss of photocatalysts in large-scale applications, 2) to control the retention time of molecules in the reactor, and 3) to continuously separate catalysts and products. Other advantages of PMRs include improved process efficiency and stability, and reduced cost by reusing photocatalysts [16].”

Q 3. What are other widely used polymeric materials for water purification?
A 3. The content of polymeric materials for water purification was supplemented on line 113 on page 3.

   “In addition to PVA, other MF and UF membrane polymers include PAN, polyether sulfone (PES), polysulfone (PSF), polypropylene (PP), polytetrafluoroethylene (PTFE), poly (vinylidene fluoride) (PVDF), and sulfonated PSF and PES. These polymer membranes show excellent selectivity, stability, and permeability during water treatment. PES and PSF membranes are not only the most used materials for UF membranes, but also widely used for NF and RO composite membranes. In addition, PP and PVDF are also widely used materials for MF membranes [10,12].”

Q 4. The Introduction section should be improved. Some of the work in literature should be discussed in details especially the developments of various membranes for water treatment process.

A 4. The contents of some of the work in literature were supplemented in line 50 on page 2 according to the reviewer's point.

“The properties and effects of several MF membranes produced for water treatment have been characterized. In 1997, Mueller et al. [8] reported that a ceramic MF membrane using polyacrylonitrile (PAN) achieved 99% purity in the treated water. Moreover, Zhong et al. [9] developed a next-generation ceramic MF membrane using zirconia (ZrO2), which has been proven to be more effective in water treatment. In addition, according to previous MF membrane studies, inorganic membranes such as ceramic membranes were more effective in reducing fouling via pre-treatment stages such as flocculation [10].”

Q 5. In Table 1, please unify the pressure units. The temperature unit is incorrect. I feel like that these conditions are not necessary.

A 5. Table 1 was removed according to the reviewer's point. Instead, the synthesis method is described in detail in section 2.2

Q 6. Did you use knife coating for the film to control the membrane thickness?
A 6. To make the film thickness constant, a solvent casting method was used. When casting the solvent several times by hand, a film having a constant thickness could be easily obtained. See page 4, 154 th line.

“The resulting PAA/PVA solution was evenly spread on a clean glass plate and cast, and then, the solvent was slowly removed while stabilizing PAA for 2 h at 50 °C in a vacuum oven.”

Q 7. In Figure 1, the unit of wavenumber should be cm-1. Figure 2 should be placed in Section 3.1.

A 7. According to the reviewer's point, the wavelength units in Figure 1 were modified and Figure 2 moved to section 3.1

Q 8. I suggest the authors to use the curves in the figures with different colors. For Figure 4, it is so hard to distinguish these lines. Why did the blended film containing 5 wt% PVA show the best thermal stability in Figure 4b?

A 8. There were some mistakes depending on the color, so we corrected it. See Figure 4 (b).

Q 9. Line 245, the PBO formation process should be listed as a scheme.
A 9. On page 9, the PBO formation process was newly created with Scheme 2

Q 10. Figure 5 is in a very low quality.
A 10. The quality of Figure 5 has been improved to make it more visible

Q 11. Figures 10 and 11 are useless if all the films are transparency.
A 11. Figures 10 and 11 were removed as a reviewer pointed out. Instead, a sufficient explanation was supplemented in section 3.4.

Q 12. The conclusions should be shortened.

A 12. The conclusion part was reduced according to the reviewer's point. See Conclusions.

Q 13. The English writing should be improved significantly.

A 13. According to the reviewer's point of view, the English of this paper was re-revised through a specialized institution called editage®

Q 14. Please improve the reference format in the revision.

A 14. The reference format was revised according to the reviewer's point.

I hope this revision is satisfactory for your further process. 

Sincerely yours,

Jin-Hae Chang

Professor 

Kumoh Nat'l Inst. of Tech.

Reviewer 2 Report

Two types of colorless and transparent polyimide (CPI) membranes were synthesized using 6FAm and 6FAm-OH monomers by blending of poly(vinyl alcohol) (PVA) into the corresponding poly(amic acid) (PAA) precursors. The thermal properties, morphology and optical transparency of these membranes were determined. It is a common method to use PVA as pore-forming agents to fabricate polymer membranes. Actually, PI membranes for microfiltration could be applied in special environment such as under high temperature. However, according to the following issues, I recommend that the paper could not be accepted at this stage.

  1. The first issue I concerned is the purpose for using colorless and transparent polyimide. In the paper I could not figure out why the authors need to utilize colorless PI, what benefice could be introduced to membranes by using colorless PIs? How about using Kapton film?
  2. As pore-forming agent, PVA was completely removed from the CPI/PVA blend film to form the porous CPI membrane. From this point of view, the effect of PVA concentration should be the pore size, the distribution of pores and the number of pores formed. However, there was no relevant results and discussions on such effect. Meanwhile, the investigations on the effect of PVA on thermal and optical properties did not make any sense, as there was no PVA at all in the CPI membrane. In addition, would the pore size has any impact on the mechanical properties of the CPI membranes?
  3. The authors did not provide the explanation why the pore size increased along with the increasing of PVA concentration. Would the PVA components aggregate at high concentration? Then how about the change of the number of pores with the PVA concentration variations?
  4. According to the introduction, the authors should give an idea on what kind of microfiltration would these membranes apply for?
  5. Page 2 line 59, “PI is dark brown due to the chain transfer complex (CT-complex)”. This should be “charge transfer complex”.

Author Response

Dear Editor

This is my response to your comments regarding our paper “Syntheses of Colorless and Transparent Polyimide Membranes for Microfiltration” (polymers-850893) in polymers.

Thank you very much for the referee's comments. I have carefully revised the manuscript following the comments of the referee.

Reviewer-2

Q 1. The first issue I concerned is the purpose for using colorless and transparent polyimide. In the paper I could not figure out why the authors need to utilize colorless PI, what benefice could be introduced to membranes by using colorless PIs? How about using Kapton film?

A 1. According to the reviewer's point of view, the advantages of CPI were supplemented. See page 3, 98th line.

   “As described above, the main chain of CPI is a kinked structure and includes an asymmetrical substituent. Thus, the processing temperature is lower and the membrane manufacturing process is superior to that of conventional PI. In addition, conventional PI is dark yellow or brown, so it is mainly used as an interior material. Although its thermal stability is slightly inferior to conventional PI, CPI is colorless and transparent. Therefore, it can be used for interior materials and also for exterior materials that require optical transparency.”

This paper focuses on the new membrane using colorless transparent PI. Therefore, no consideration was given to the brown Kapton film. Also, Kapton cannot be used because it is not blended with PVA

Q 2. As pore-forming agent, PVA was completely removed from the CPI/PVA blend film to form the porous CPI membrane. From this point of view, the effect of PVA concentration should be the pore size, the distribution of pores and the number of pores formed. However, there was no relevant results and discussions on such effect. Meanwhile, the investigations on the effect of PVA on thermal and optical properties did not make any sense, as there was no PVA at all in the CPI membrane. In addition, would the pore size has any impact on the mechanical properties of the CPI membranes?

A 2. According to the reviewer's point, the points were supplemented in line 267 on page 8 and in line 417 on page 13..

 “For the two CPI membrane series, no transition temperatures (Tg, Tm, and TDi) corresponding to PVA was observed, and the thermal properties of each series were almost the same regardless of the PVA content; this indicates that PVA was completely removed from the CPI blend and a pure CPI membrane was obtained.”

     “Since the PVA component of the CPI membrane was completely removed, it was not possible to determine the effect of the PVA content on the optical transparency in the two membranes series, as demonstrated with the thermal properties of the films.”

       Since the thermal properties and optical transparency of the CPI membrane are the same regardless of the PVA content, it is expected that the mechanical properties of the CPI membrane will be the same regardless of the PVA content

Q 3. The authors did not provide the explanation why the pore size increased along with the increasing of PVA concentration. Would the PVA components aggregate at high concentration? Then how about the change of the number of pores with the PVA concentration variations?

A 3. According to the reviewer's point, the point was supplemented in line 338 on page 10.

“As the PVA content of the CPI blend increased from 0.5 to 5.0 wt%, PVA agglomerated to increase the particle size. When these particles are dissolved in water and removed, the pore size in the CPI membrane correspondingly increased. The shape of the membrane pores depending on the concentration of PVA is also shown in Figure 6.

Q 4. According to the introduction, the authors should give an idea on what kind of microfiltration would these membranes apply for?

A 4. According to the reviewer's point, the point was supplemented in line 437 on page 13.

“In conclusion, these CPI membrane films are expected to be useful as high-functional polymeric materials as well as in the field of filtration due to their superior thermal property and optical transparency as compared to that of general-purpose engineering polymers. Specifically, in the case of 6FAm-OH PI, because of its hydrophilicity, the pore size can be easily adjusted. Therefore, it is expected that 6FAm-OH PI can be used for MF membranes that require optical transparency.”

Q 5. Page 2 line 59, “PI is dark brown due to the chain transfer complex (CT-complex)”. This should be “charge transfer complex”.

A 5. According to the reviewer's point, the chain was replaced with charge in line 91 on page 2.

I hope this revision is satisfactory for your further process. 

Sincerely yours,

Jin-Hae Chang

Professor 

Kumoh Nat'l Inst. of Tech.

 Round 2

Reviewer 1 Report

In Table 1, please revise "0 (pure P".

In these SEM photos, it is hard to see the scales of the figures.

Author Response

Dear Editor

This is our response to reviewer’s your comments regarding our paper “Syntheses of Colorless and Transparent Polyimide Membranes for Microfiltration” (polymers-850893) in polymers.

Thank you very much for the reviewer's comments. I have carefully revised the manuscript following the comments of the reviewer.

Open Reviewer-1

Comments and Suggestions for Authors

Q-1. In Table 1, please revise "0 (pure P".

A-1. I didn't understand the question. Very sorry.

Q-2. In these SEM photos, it is hard to see the scales of the figures.

A-2. As the reviewer pointed out, the scale bar was added to the picture to make the picture appear well. See Figures 6-8.

I hope this revision is satisfactory for your further process. 

Sincerely yours,

Jin-Hae Chang.

Professor 

Kumoh Nat'l Inst. of Tech.

Reviewer 2 Report

  1. The potential application of these CPI membranes is microfiltration, which would require to be used under high pressure. Therefore, the mechanical property of these CPI membranes is much more important than their thermal and optical properties. The formation of holes in the CPI membrane can possibly reduce the mechanical strength. It is not convincing that the authors only gave a deduction on the effect of pore size on mechanical properties. Therefore, the authors need to give experimental date to compare the mechanical properties between the pure CPI film and the porous CPI membrane.
  2. The authors need to clarify what is the real role of PVA in the CPI membrane? After I read this draft, I only found PVA is just a hole-forming reagent. It was completely removed from the CPI membrane. So, the description on PVA in the introduction section (lines 101–115) is not proper, need to be reworded.
  3. Please double-check Figure 9. It was not in the same order compared to the first version (see (a) and (b)).

Author Response

Dear Editor

This is our response to reviewer’s your comments regarding our paper “Syntheses of Colorless and Transparent Polyimide Membranes for Microfiltration” (polymers-850893) in polymers.

Thank you very much for the reviewer's comments. I have carefully revised the manuscript following the comments of the reviewer.

Open Reviewer-2

Comments and Suggestions for Authors

Q-1. The potential application of these CPI membranes is microfiltration, which would require to be used under high pressure. Therefore, the mechanical property of these CPI membranes is much more important than their thermal and optical properties. The formation of holes in the CPI membrane can possibly reduce the mechanical strength. It is not convincing that the authors only gave a deduction on the effect of pore size on mechanical properties. Therefore, the authors need to give experimental date to compare the mechanical properties between the pure CPI film and the porous CPI membrane.

A-1. We also fully agreed with your opinion and tried to measure the mechanical properties of the membrane film. The mechanical properties of pure PI were obtained, but on the contrary, in the case of the PI blend membrane, a constant value was not obtained despite several attempts.

     Most of the membrane films were torn during the measurement and did not show constant mechanical properties. This is thought to be due to the small pores present in the membrane film and it became increasingly difficult to measure mechanical properties as the PVA increased to 5 wt%. This is because the pore size increases as the amount of PVA increases in the PI blend membrane.

Table. Tensile properties of CPI membrane films with various PVA contents.

PVA in PI (wt%)

6FAm PI

6FAm-OH PI

Ult. Str.

(MPa)

Ini. Mod.

(GPa)

E.B. a

(%)

Ult. Str.

(MPa)

Ini. Mod.

(GPa)

E.B.

(%)

0 (pure PI)

37

2.00

2

55

3.28

2

0.5

1.0

Not observed

2.0

5.0

a Elongation percent at break.

Q-2. The authors need to clarify what is the real role of PVA in the CPI membrane? After I read this draft, I only found PVA is just a hole-forming reagent. It was completely removed from the CPI membrane. So, the description on PVA in the introduction section (lines 101–115) is not proper, need to be reworded.

A-2. According to the reviewer's point of view, we have modified the content of the PVA. See line 104 on page 3.

“Poly(vinyl alcohol) (PVA) is a water-soluble polymer extensively used in paper coating, textile sizing, and production of flexible water-soluble packaging films [27]. Such applications have stimulated interest in improving the mechanical, thermal, and permeability properties of thin film composite (TFC) membrane [11, 28-30]. Since PVA exhibits strong water solubility, it is possible to make pores of uniform size when preparing a TFC membrane in aqueous solution. This result allows the membrane to maintain uniform properties.”

Q-3. Please double-check Figure 9. It was not in the same order compared to the first version (see (a) and (b)).

A-3. Thank you for pointing out in detail. In Figure 9 (a) and (b), PVA in 6FAm and PVA in 6FAm-OH were modified to PVA in 6FAm PI and PVA in 6FAm-OH PI, respectively. See Figure 9.

I hope this revision is satisfactory for your further process. 

Sincerely yours,

Jin-Hae Chang.

Professor 

Kumoh Nat'l Inst. of Tech.

Round 3

Reviewer 2 Report

Accept in present form.